# Metrology of the Solar Spectral Irradiance at the Top Of Atmosphere in the Near Infrared Measured at Mauna Loa Observatory: The PYR-ILIOS campaign

Nuno Pereira[1], David Bolsée[1], Peter Sperfeld[2], Sven Pape[2], Dominique Sluse[1], and Gaël Cessateur[1]

[1]BIRA-IASB, 3 Ringlaan, 1180 Brussels, Belgium
[2]Physikalisch-Technische Bundesanstalt, Braunschweig, Germany

*Correspondence to:* nuno.pereira@aeronomie.be

**Abstract.** The near infrared (NIR) part of the solar spectrum is of prime importance for the solar physics and climatology, directly intervening in the Earth's radiation budget. Despite its major role, available solar spectral irradiance (SSI) NIR datasets, space-borne or ground based, present discrepancies caused by instrumental or methodological reasons. We present new results obtained from the PYR-ILIOS SSI NIR ground-based campaign, which is a replication of the previous IRSPERAD campaign which took place in 2011 at the Izaña Observatory (IZO). We used the same instrument and primary calibration source of spectral irradiance. A new site was chosen for PYR-ILIOS: the Mauna-Loa observatory in Hawaii (3397 m asl), approximately 1000 m higher than IZO. Relatively to IRSPERAD, the methodology of monitoring the traceability to the primary calibration source was improved. The results as well as a detailed error budget are presented. We demonstrate that the most recent results, from PYR-ILIOS and other space-borne and ground-based experiments show an NIR SSI lower than previous reference spectrum, ATLAS3, for wavelengths above $1.6\mu m$.

## 1 Introduction

An accurate knowledge of the solar spectral irradiance (SSI) remains central to the study of the climate on Earth. The variability in the ultraviolet (UV) part of the spectrum and its influence on climate via the mechanisms of solar-terrestrial interactions, simulated by chemistry-climate models (CCM) (Gray et al., 2010; Ermolli et al., 2013), makes up for the most of the research in SSI measurements. Despite of its extremely low variability, $< 0.05\%$ over a solar cycle, (Lean, 1991; Harder et al., 2009) the near-infrared (NIR) part of the spectrum plays a major role in the Earth's radiative budget due to its quasi-total absorption by water vapour (Collins et al., 2006). The determination of its absolute level remains challenging (Meftah et al., 2017): the measurement of the top of atmosphere (TOA) SSI started nearly 50 years ago and evolved both with ground-based and space borne instruments and a consensus on the absolute level in the NIR part is still to be achieved (Bolsée et al., 2014; Hilbig et al., 2018).

Aircraft borne instrumentation at an altitude of 12 km provided the first TOA SSI measurements dataset in 1969 (Arvesen et al., 1969) with an onboard standard of spectral irradiance.

Several ground-based measurements campaigns in the UV, Visible and NIR, have been conducted from the top two mountain-top reference sites afterwards:

- Izaña Atmospheric Observatory (IZO): IRSPERAD (Bolsée et al., 2014) with a NIR (0.6 $\mu m$ - 2.3$\mu m$) spectroradiometer and the QASUMEFTS (Gröbner et al., 2017) instrument providing a high-resolution UV spectrum; both were calibrated against the Physikalisch-Technische Bundesanstalt (PTB) BB3200pg blackbody (Sapritsky et al., 1997; Sperfeld et al., 1998a, 2000);

- Mauna Loa Observatory (MLO): Shaw (1982) with a 10-channel (UV, Visible and NIR) filter radiometer; Gröbner and Kerr (2001) with a double Brewer spectrophotometer measuring in the 300 $nm$ - 355$nm$; Kindel et al. (2001) provided TOA SSI in the range 350 $nm$ to 2500 $nm$, measured with a spectroradiometer; all these measurement campaigns used different types of 1000$W$ lamps, traceable to National Institute of Standards and Technology (NIST) standards as calibration sources.

Finally, the CAVIAR (Menang et al., 2013) and CAVIAR2 (Elsey et al., 2017) spectra, obtained with an infrared Fourier spectrometer (FTIR) calibrated against National Physical Laboratory (NPL) standards at the United Kingdom Met Observation site in Camborne, in the range 1 $\mu m$ - 2.5$\mu m$.

TOA SSI values from all referred ground-based campaigns were obtained with the Langley-plot technique that permits the extrapolation to the TOA irradiance in criteriously chosen atmospheric windows (Section 2.2). The monitoring of the absolute spectral calibrations is secured by comparisons with relative stable secondary standards. The reliability of the traceability to primary irradiance standards is an advantage for ground based measurement. Performing these measurements based on world reference sites for the determination of TOA physical quantities, such as IZO and MLO, on often occurring pristine day conditions ensures a high accuracy of the TOA extrapolations (Shaw, 1975, 1976; Kiedron and Michalsky, 2016; Toledano et al., 2018).

On the other hand, space borne SSI measurements covering the NIR range, started in the 1990s, however limited to wavelengths shorter than 2.4 $\mu m$. From the SOLSPEC instrument family, the instrument SOSP (SOlar SPectrum Instrument) on board EURECA (Thuillier et al., 1981) that pionereed the space-borne NIR absolute solar spectroscopy released the ATLAS3 reference spectrum (Thuillier et al., 2003). An upgraded version of SOLSPEC instrument, SOLAR/SOLSPEC, including a fully refurbished NIR channel, readout electronics and extended wavelength range up to 3 $\mu m$ of SOLSPEC flew from 2008-2017 on board the International Space Station (ISS) (Thuillier et al., 2009), releasing the SOLAR2 (Bolsée, 2012; Thuillier et al., 2013) and SOLAR-ISS(IR) (Meftah et al., 2017); SOLSPEC is the space borne instrument that measured farther the SSI in the NIR. The instrument providing the longest time series of SSI measurements in the NIR is the SIM (Spectral Irradiance Monitor) prism spectrometer on SORCE (Solar Radiation and Climate Experiment) launched in 2003 (Harder et al., 2000a, 2005) and still on orbit but with scarce operational time, due to the end of battery life. Another instrument contributing to NIR SSI measurements is SCIAMACHY (Scanning Imaging Absorption Spectrometer for Atmospheric Chartography) (Noël et al., 1998; Burrows et al., 1995), a remote sensing spectrometer adapted to measure SSI. The latest data release is SCIAMACHY V9 (Hilbig et al., 2018).

All above referred NIR datasets reasonably agree up to $1.3\mu m$. When comparing SORCE and ATLAS3, the difference between both do not exceed $2\%$ in the NIR range which is a consequence of SORCE being scaled up to ATLAS3, due to incompatibilities of fractional TSI (Total Solar Irradiance) between both datasets (Harder et al., 2010).

At $1.6\mu m$, corresponding to the minimum opacity value of the solar photosphere, differences up to $8\%$ (reaching $10\%$ at $2.2\mu m$) were observed between ATLAS3 with respect to SOLAR2 (Thuillier et al., 2013). This bias motivated the development of new ground-based instrumentation measuring the SSI NIR: CAVIAR and IRSPERAD (Bolsée et al., 2014; Menang et al., 2013). The data of both experiments confirmed this bias, both showing a level closer to that of SOLAR2. Posteriorly, SOLSPEC and SCIAMACHY data reprocessing tend to intermediate values between ATLAS3 and SOLAR2 (Meftah et al., 2017; Hilbig et al., 2018).

In this paper we present a re-run of the IRSPERAD experiment, named PYR-ILIOS, carried out in July 2016. While still using the Langley plot technique and calibration against the PTB blackbody, this new experiment differs from IRSPERAD in three aspects: first, the observation site is MLO instead of IZO; second, possible sources of systematic uncertainties have been identified and fixed (see Sec. 2.1); third, the traceability of the calibration to the primary standard was improved (see Sec. 2.6). A detailed estimation of the uncertainty budget will be presented in Sec. 3, followed by the presentation of the obtained spectrum and its comparison with space borne and ground-based spectra described in this chapter along with a discussion on the status of the NIR SSI measurement.

## 2 Methods

### 2.1 Instrumentation

The core of the direct Sun measurement instrumentation is a Bentham NIR spectrometer: it consists of a double monochromator placed inside a thermally stabilized container, with light detection by a PbS cell. A fiber optic guides the Sun light between the entrance slit of the spectrometer and the diffusor of a $7.2°$ field-of-view (FOV) Sun light-collecting optics (telescope). The telescope is connected to an EKO Sun tracker that provides a tracking accuracy[1] of $0.01°$. The working wavelength range is from $0.6\ \mu m$ to $2.3\ \mu m$, with a nominal $10\ nm$ bandpass. The instrument characteristics are given in depth in Bolsée et al. (2014) and have remained unchanged since. No modifications have been made neither to the telescope nor to the spectrometer. Nevertheless, a factory defect in the assemblage of the components was detected and rectified: the lens focusing the light collected in the optic fiber into the spectrometer entrance slit was properly fixed into its barrel support for the PYR-ILIOS campaign, which wasn't previously the case for the IRSPERAD campaign at IZO. Another change relative to the IRSPERAD campaign was that the thermally stabilized spectrometer container was placed indoors in a thermally stabilized environment, which reduced thermal stress due to outdoor exposure and improved the stability of the spectrometer's response.

---

[1]https://eko-eu.com/products/solar-energy/sun-trackers/str-22g-sun-trackers

## 2.2 Langley-plot method

The wavelength-dependent direct transmitted solar irradiance in the atmosphere is described by the Beer-Bouguer-Lambert (BBL) law. For spectral regions where molecular absorption is negligible and only Rayleigh and aerosol scattering are present BBL law is written in the form:

$$E\left(\lambda\right) = E_0\left(\lambda\right) D^{-2} \exp\left[-m_R\left(\theta\right)\tau_R\left(\lambda\right) - m_A\left(\theta\right)\tau_A\left(\lambda\right)\right], \tag{1}$$

where $E_0$ is the irradiance at the top of atmosphere (TOA), $m$ is the air mass factor (AMF) as a function of the solar zenith angle (SZA) $\theta$, and $\tau$ is the optical depth that depends on $\lambda$. $D$, is the ratio between the Earth-Sun distance at the moment of the measurement and the mean Earth-Sun distance; subscripts $R$ and $A$ stand for Rayleigh and aerosol, respectively. Because the aerosol vertical profile over the measurement site at the moment of the measurement is unknown, aerosol AMF is approximated to Rayleigh AMF (Schmid and Wehrli, 1995); considering $m_A \approx m_R \approx m$, defining $\tau = \tau_R + \tau_A$ and taking the logarithm of Eq. 1, it can be re-written as:

$$\log\left[E\left(\lambda\right)\right] = \log\left[E_0\left(\lambda\right) D^{-2}\right] - m\left(\theta\right)\tau\left(\lambda\right). \tag{2}$$

Provided that $\tau\left(\lambda\right)$ remains constant for a series of measurements of $E_0\left(\lambda\right)$ taken over a given range of $m\left(\theta\right)$ (spreading over a half day), the TOA value of $E_0\left(\lambda\right)$ is thus the intercept at the origin ($m = 0$) of the least squares regression to the data series $E\left(\lambda\right)$ as a function of $m(\theta)$.

Solar Zenithal Angles (SZA) are calculated with NOAA Solar Position Calculators[2] that implement Meeus (1998) algorithms and subsequently corrected for atmospheric refraction effects according to Bennett (1982). AMF are calculated using the Kasten and Young algorithm (Kasten and Young, 1989).

## 2.3 Atmospheric windows

The wavelength domains for which the Langley-plot method described in Sec. 2.2 is valid, i.e. atmospheric windows, were determined through model using a procedure developed in Kindel et al. (2001) and also used in Bolsée et al. (2014): using a TOA reference spectrum as input, MODTRAN (MODerate resolution atmospheric TRANsmission) (Berk et al., 2014) RTM (Radiative Transfer Model) was used to simulate irradiances measured at ground, as a function of the measurement site parameters, for a series of AMFs. The Langley-plot method was applied to these simulated irradiances and the wavelengths for which the synthetic $E_0$ recreated the input TOA within $0.5\%$ were kept as valuable wavelengths for the Langley-plot; these set of wavelengths were grouped in contiguous windows called atmospheric windows.

---

[2]https://www.esrl.noaa.gov/gmd/grad/solcalc/index.html

## 2.4 Absolute Calibration

The absolute calibration was performed against a primary standard of spectral irradiance, the BB3200pg blackbody of the PTB. It has been extensively described in Sapritsky et al. (1997) and Sperfeld et al. (1998a, 2000). The spectral irradiance equation describing the black body emission is calculated using Planck's law:

$$E_{\mathrm{BB}}(\lambda) = \varepsilon_{\mathrm{BB}} \frac{A_{\mathrm{BB}}}{d_{\mathrm{BB}}^2} \frac{c_1}{n^2 . \lambda^5} \frac{1}{\exp\left(\frac{c_2}{n_\lambda . \lambda . T_{\mathrm{BB}}}\right) - 1},$$

(3)

where $\varepsilon_{\mathrm{BB}}$ and $A_{\mathrm{BB}}$ stand, respectively, for the effective emissivity and the aperture of the BB3200pg, $d_{\mathrm{BB}}$ for the distance between the blackbody aperture and the optic center of the telescope, $n$ for the refractive index of air, $c_1$ and $c_2$ are the first and second radiation constants.

The fundamental parameter, the temperature of the cavity $T_{BB}$ is known with a standard uncertainty of 0.5 K ($\sim 0.02\%$ for a nominal temperature of 3000K) with a drift lower than $0.5$ Kh$^{-1}$ (Friedrich et al., 1995; Werner et al., 2000; Taubert et al., 2003). The uncertainties on $\varepsilon_{\mathrm{BB}}$ and $A_{\mathrm{BB}}$ are $1 \times 10^{-4}$ (0.01%) and 0.04mm (0.03%), respectively (Woolliams et al., 2006). The distance between the blackbody aperture and the telescope optical active surface, the diffuser $d_{BB}$ is the sum of two distances: $d_{BB} = d_s + d_T$, where $d_S$ is the distance between the black body and the first optical surface of the telescope, the quartz plate, and $d_T$ is the distance between the quartz plate and the diffuser. The uncertainties on $d_s$ and $d_T$ are 0.05 mm (Woolliams et al., 2006) and 0.5 mm, respectively; the combined uncertainty on $d_{BB}$ is of 0.5mm, $0.04\%$ at the nominal distance of 1384.05 mm.

The absolute calibration coefficient $R$, that converts the spectrometer signal into irradiance is given by Eq. 4:

$$R(\lambda) = \frac{E_{\mathrm{BB}}(\lambda, T)}{S_{\mathrm{BB}}(\lambda)},$$

(4)

With $S_{\mathrm{BB}}$ being the signal recorded by the spectrometer and $E_{\mathrm{BB}}$, the emission of the blackbody, given by Eq. 3. During the calibration campaign at PTB, two different temperatures set points, 3016.5K and 2847.6K, were used to build the response curve, $R_{\mathrm{BB}}$. The distance $d_{\mathrm{BB}}$ was kept fixed at 1384.05 mm so that the blackbody aperture was seen by the entrance optics with an angular extension of $0.5°$.

## 2.5 Radiometric characterization

The spectrometer was characterized at the laboratory of the Belgian Institute for Space Aeronomy (BIRA-IASB) for the uncertainty on the measured signal, the detector sensitivity to temperature and for the wavelength scale. The flat field of the detector was measured during the ground-based campaign at MLO and the linearity was verified during the calibrations at PTB laboratory:

- The flat field of the entrance optics was measured during the ground-based campaign. The telescope was angularly displaced from the normal Sun direction thanks to an angular fine tuning mechanism, for a series of angular positions

for two orthogonal directions. The agreement between both directions data curves allows to estimate an insensitivity of the signal to Solar depointing better than $0.05°$, altough a finer angular sampling would be necessary to accurately determine the angular limits of this insensitivity. Given the $0.01°$ pointing accuracy of the Sun tracker, the response of the instrument is considered to be insensitive to pointing during the campaign.

5    - The temperature sensitivity of the spectrometer was determined in laboratory (Bolsée et al., 2014). During the campaign, the spectrometer box was placed indoors with its temperature being constant within $0.1°C$, equivalent to the resolution of the temperature probe readout; no temperature correction on the signal was thus applied.

   - For the verification of the linearity of the detector, the telescope was placed at several different distances from a stable 200W lamp. The measured signal as a function of distance were successfully fitted to an inverse square law function,

10    demonstrating the detector linearity within a 2 decade dynamic range.

## 2.6   Relative Calibration

A set of 6 FEL lamps (F102, F104, F417, F418, F545, F546) were used as relative calibration standards, to monitor a possible change of response of the spectrometer during the measurement campaign. Taking as reference the lamps signal measured at the PTB (April 27 2016), $S_{Fj}^{PTB}(\lambda)$, four additional relative calibrations were performed:

15    - Immediately before the start of the measurement campaign on June 29 ($i = 1$), the signal of the 6 lamps, $S_{Fj}^{MLO1}$, was measured on site. This first MLO relative calibration was valuable to monitor the spectrometers response change between the calibration at PTB and the beginning of the field measurements. During this 2-month period that included the transportation of the equipment, a decrease of response varying between $1\%$ and $3\%$ in the 1000 nm to 2200 nm range was detected.

20    - During the 20-day measurement campaign, three relative calibrations were performed: on July 7 ($i = 2$), 14 ($i = 3$) and 19 ($i = 4$). The cumulated loss of response between June 29 and July varied from $1.5\%$ to $0.5\%$ in the $800nm$ to $1.8\mu m$ domain.

The corresponding correction factor for each relative calibration is:

$$K_i(\lambda) = \frac{1}{N} \sum_{j}^{N} \frac{S_j^{MLOi}(\lambda)}{S_j^{PTB}(\lambda)}, \tag{5}$$

25   where $N$ stands for the total number of lamps, $j$ to the lamp number and $i$ to the calibration day index. $K(\lambda)$ was obtained by linear interpolation for all days of the campaign.

## 2.7   Ground-based campaign

The PYR-ILIOS campaign took place during the first 20 days of July 2016 at the Mauna Loa Observatory (MLO) in the island of Hawaii. The MLO (19.53° N,155.58°W) is situated at 3397 m above sea level; it is the leading long-term atmospheric

monitoring facility on Earth, a primary calibration site for the AErosol Robotic NETwork (AERONET[3]) network, a global station for the Global Atmosphere Watch (GAW) of the World Meteorological Organization (WMO) and the premier site[4] for the measurement of the concentration of atmospheric carbon dioxide. It is considered a world reference site to accurately determine extraterrestrial constants via the Langley-plot method (Shaw, 1975, 1976; Kiedron and Michalsky, 2016).

## 2.8 Data selection and analysis

From the 20-day campaign, 12 high-quality half-days, all during morning time, were kept for analysis. The selection criteria were: verification of cloudless clear skies and a Langley-plot correlation coefficient $R^2 > 0.9$. The morning data of the days 2, 3, 5, 7, 8, 9, 10, 11, 13, 14, 16, 17 of July 2016 data were kept for analysis:

A subset of these selected Langley plots is shown in Fig. 1, for four different wavelengths.

## 3 Uncertainty budget

### 3.1 Uncertainty on the spectrometer signal

The raw uncertainty of a spectrometer measured signal, $S_x^{raw}$ regardless of its source, either Solar ($S_S$), blackbody ($S_{BB}$) or lamps $\left(S_{Fj}^{PTB}, S_{Fj}^{MLO}\right)$ signal, is a function of the intrinsic noise of the measured physical signal convolved by the spectrometer's transmission and detector's response. The uncertainty on a measured signal $u(S_x^{raw})$ was determined in laboratory by calculating the standard deviation for a sample of measured signals at several intensities from a 1000W stable lamp (Bolsée et al., 2014). This uncertainty is shown in Fig. A1.

Additionally, all measured signals, $S_x^{raw}(\lambda)$, are affected by an uncertainty term due to the finite bandpass of the instrument, $u(C_{\Delta\lambda})$, and the uncertainty on the determination of the true wavelength scale, $u(C_\lambda)$ (Obaton et al., 2007).

$$u(S_x(\lambda))^2 = u(S_x^{raw}(\lambda))^2 + u\left[C_\lambda(S_x^{raw}(\lambda), \delta(\lambda))\right]^2 + u\left[C_{\Delta\lambda}(S_x^{raw}(\lambda), BW)\right]^2, \tag{6}$$

where $\delta(\lambda)$ stands for the maximum deviation in the determination of the real wavelength scale of the spectrometer. $\delta(\lambda)$ was determined in laboratory by measuring the deviation between the measured and the corresponding nominal peak values of a series of well known emission rays of Xe, Ar and Kr lamps as well as of lasers and pen-ray lamps; $\delta(\lambda) < 0.2nm$ for the working wavelength range. $BW$ stands for the spectrometer bandpass of 10.63 nm, measured in laboratory.

### 3.2 Uncertainty on a calibrated direct Sun measurement

The expression for a calibrated solar measurement, $E(\lambda)$ is:

$$E(\lambda) = S_S(\lambda).R(\lambda).K(\lambda), \tag{7}$$

---

[3]https://aeronet.gsfc.nasa.gov/
[4]https://www.esrl.noaa.gov/gmd/obop/mlo/programs/esrl/co2/co2.html

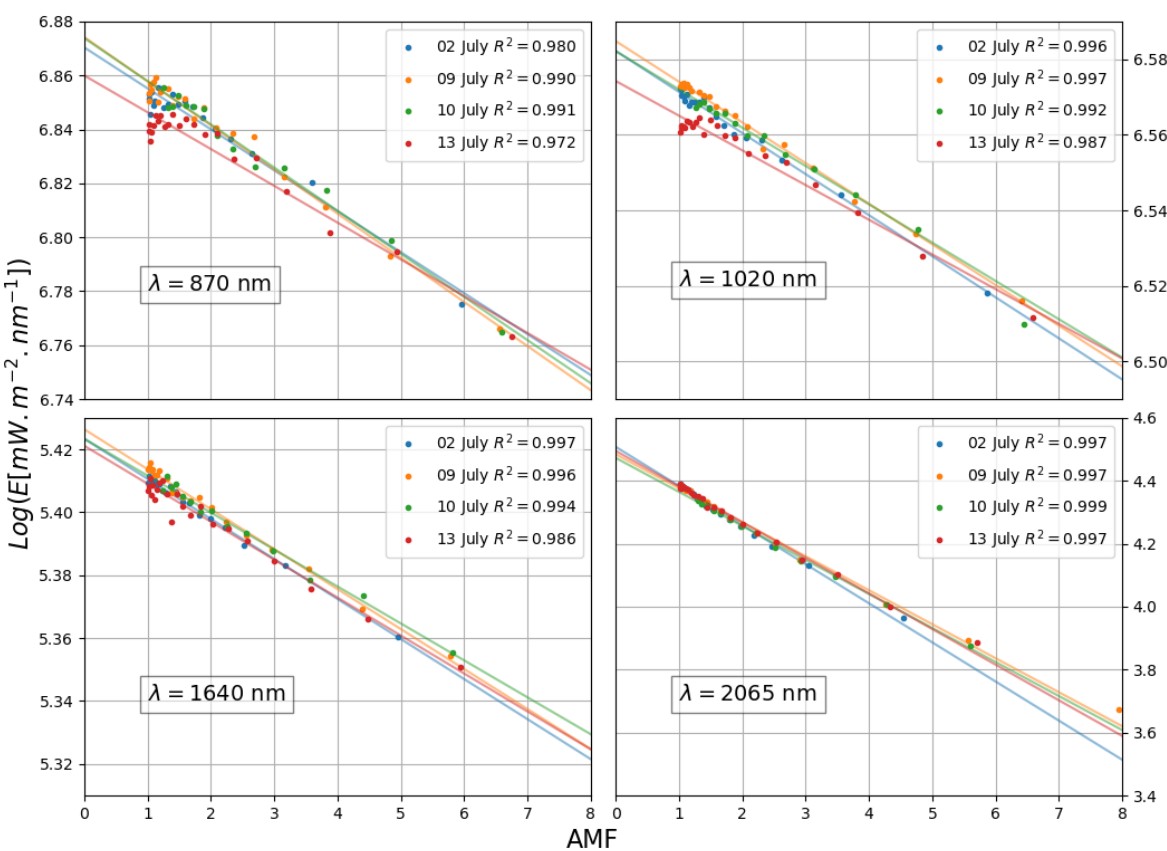

**Figure 1.** Measured irradiance and respective Langley plot fits for the 3 AERONET wavelenghts, 870 nm, 1020 nm and 1640 nm and 2065 nm, shown for the morning data of 2, 9, 10 and 13 July 2016.

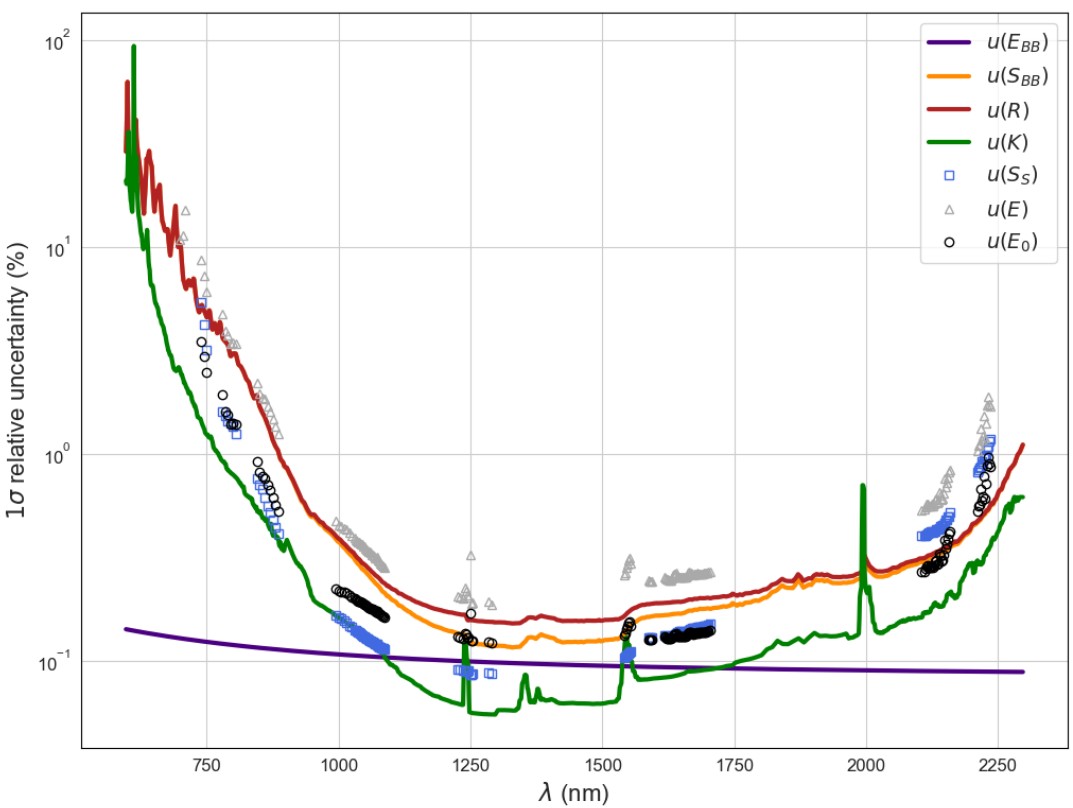

**Figure 2.** Individual uncertainties contributing to the combined uncertainty in the TOA SSI, $u(E_0)$. Blackbody ($u(E_{BB})$, $u(S_{BB})$ and $u(R)$) and Lamps ($u(K)$) associated quantities are plotted for the full wavelength working range, while solar measurement associated quantities ($u(S_s)$, $u(E)$ and $u(E_0)$) are plotted in the atmospheric windows wavelengths.

With $S_S(\lambda)$, $R(\lambda)$ and $K(\lambda)$ being expressed by Eq. 6, 4 and 5, respectively. The uncertainties associated to the factors in Eq. 7 were calculated using the Law of Propagation of Uncertainties (LPU) and are represented in Fig. 2. The similarity of shapes of the curves of the individual uncertainties reflects the convolution of the measured signals by the spectrometer's response. The largest contribution to the calibrated solar signal comes from the uncertainty on the absolute calibration which is dominated by the uncertainty $u(S_{BB})$ of the measured signal, $S_{BB}$, of the blackbody, whereas the uncertainty on the emission of the blackbody, $u(E_{BB})$, is known within $0.2\%$ for the totality of the wavelength range.

## 3.3 Uncertainty on the determination of the TOA irradiance

The uncertainty in the determination of the TOA irradiance via the Langley plot method, $u(E_0^{LP})$ corresponds to the uncertainty on the determination of the intercept at origin, $P_0$, when applying a linear regression on Eq. 2. The uncertainty on the measured $E$ in the Langley-plot method logarithmic space, $u(log(E))$ and the uncertainty in the $u(E_0^{LP})$ irradiance value are given by:

$$u^2(log(E)) = \left(\frac{\partial log(E)}{\partial E}\right)^2 . u^2(E) = \left(\frac{u(E)}{E}\right)^2 \tag{8}$$

$$u^2(E_0^{LP}) = \left(\frac{\partial exp(P_0)}{\partial P_0}\right)^2 . u^2(P) = exp(P_0)^2 . u^2(P_0), \tag{9}$$

where, $E_0^{LP} = exp(P_0)$ gives the irradiance TOA value. The uncertainty in $P_0$ was estimated by 2 independent methods.

- A Monte Carlo method: given a measured Langley plot dataset consisting of $(m_i, log(E_i))$ points, a new synthetic dataset $(m_i^*, log(E_i^*))$ is created where each $log(E_i^*)$ is affected by a random normal distributed quantity, with a standard uncertainty given by Eq.8 and each $m_i$ is affected by an uncertainy defined in Sec. 3.5. The standard deviation in the distribution of the $N >> 1$ retrieved $P_0$ values, corresponds to $u(P_0)$, with $u(E_0^{LP})$ given by Eq. 9.

- The weighted total least-squares (WTLS) algorithm developed Krystek and Anton (2007) was used. It computes the uncertainy in the determination for both linear regression parameters using the uncertainties on the measured quantities as inputs, i.e. the uncertainties on $E$ (Sec. 3.2) and AMF (Sec. 3.5).

The uncertainty on the determination of the TOA irradiance, $u(E_0)$, matches perfectly for both methods; it is below $1\%$ for the central wavelength range of $0.9\mu m$ to $2.2\mu m$. Figure 2 shows the contribution of all the uncertainty terms detailed in Sec. 3. In Table 1 a listing of the uncertainty types and values at key wavelengths is presented.

## 3.4 Quantification of the circumsolar radiation

An ideal sun-collecting optic device should ideally have an acceptance angle equal to that of the solar disk seen on earth, $\sim 0.5°$. In practice the FOV of is much larger than $0.5°$ such that Sun sky-scattered radiation enters the FOV of the sun-collecting optics, affecting the direct normal Sun measurement. Circumsolar radiation is strongly dependent on aerosols size

and its abundance, increasing with AMF and decreasing with wavelength due to Rayleigh scattering, (Blanc et al., 2014). The estimation of circumsolar radiation was done with the aid of the LibRadtran (Mayer and Kylling, 2005) RTM. LibRadtran computes the radiance field of the Sun sky-scattered radiation. The integral of this radiance field over the solid angle of the acceptance cone of the entrance optics is the amount of circumsolar irradiance (CSI) measured by the spectrometer in excess of the normal direct Sun irradiance (DNI) (Gueymard, 2001). For standard clear-sky atmospheric conditions observed at MLO and for typical aerosol charges values measured during the mission, the quantification of CSI is shown in Fig. A2. Given the uncertainty budget, the impact of the circumsolar radiation can be considered negligible.

## 3.5 Estimation of Air Mass Factors uncertainty

As referred in Sec. 2.2 the absence of knowledge of the vertical profile of the relevant species, namely aerosols, is a limiting factor for accuratelly calculating the AMF. The uncertainty in the AMF calculation is based on Schmid and Wehrli (1995) approach who considered that $m_A$, due to the presence of stratospheric aerosols, could take the form $m_A = k_1.m_R + k_2.m_{O_3}$, with $k_1 + k_2 = 1$ and $m_{O_3}$ standing for the ozone air mass. Assuming a rectangular distribution of $m_A$ delimited by $k_1 = 1$ and a $k_1 = 0.2$, the standard deviation of $m_A$ can be calculated as: $u(m_A) = \mid m_A(k_1 = 1) - m_A(k_1 = 0.2) \mid . \frac{1}{2\sqrt{3}}$ to be used as input for the determination of the Langley plot parameters uncertainty (Sec. 3.3).

## 3.6 Langley plot sensitivity to aerosol daily variation

The possible bias introduced at the Langley plot's intercept at origin by a realistic non constant aerosol concentration during the measurement was estimated considering a measured AOD profile. For a given measured Langley plot consisting of $(m_i, log(E_i))$ and regression parameters $E_0^{LP}$ and $\tau$, a synthetic Langley plot $(m_i, log(E_i^*))$ is determined. The synthetic $E_i^*$ are calculated with the expression $E_i^* = E_0^{LP}.\exp(-m_i\tau_i^*)$ where $\tau_i^* = \tau_i^*(\lambda, t_i) = \tau_{AOD}(\lambda, t_i) + \tau_R(\lambda, t_i)$; $\tau_{AOD}(\lambda, t_i)$ stands for the real diurnal aerosol optical depth profile measured with AERONET (available at $\lambda = 870nm$, $1020nm$, $1640nm$) and $\tau_R(\lambda, t_i)$ the Rayleigh optical depth calculated according to Bodhaine et al. (1999). This bias at the intercept at origin, expressed as ratio, $E_0^{LP}/E_0^*$, averaged over the selected days is of $-0.2\%$, $+0.4\%$ and $+0.1\%$ for 870 nm, 1020 nm and 1640 nm, respectively. The signal of the bias replicates the signal of the AOD morning trend measured at MLO and the larger negative bias at 1020 nm relative to 1640 nm is due to the more pronounced AOD negative trend at 1640 nm. Assuming that the interval $\mid E_0^{LP} - E_0^* \mid$ comprises the true value of the intercept at origin, $E_0$, within a rectangular distribution, the corresponding uncertainty $u(E_0^{AOD}) = \frac{\mid E_0^{LP} - E_0^* \mid}{2\sqrt{3}}$ amounts to $0.06\%$ at 870 nm and 1640 nm and $0.1\%$ at 1020 nm which is added quadratically to the uncertainty on $E_0^{LP}$ (Sec. 3.3) to determine the uncertainty on $E_0$. $u(E_0^{AOD})$ is interpolated linearly to the working wavelength range.

**Table 1.** List of relative uncertainties terms expressed in percentage. The coverage factor is $k = 1$ for all terms. A, B stand respectively for type A and type B uncertainties, while C stands for combined uncertainty according to GUM (2008). $u(C_\lambda)$ and $u(C_{\Delta\lambda})$ are calculated for a Solar signal. The prefix $u$, for uncertainty, is omitted for each term of the first row, for sake of clearness.

| | $AMF$ | | $T_{BB}$ | $A_{BB}$ | $\epsilon_{BB}$ | $d_{BB}$ | $E_{BB}$ | $S_{BB}$ | $C_\lambda$ | $C_{\Delta\lambda}$ | $S_S$ | $K$ | $R$ | $E$ | $E_0^{LP}$ | $E_0^{AOD}$ | $E_0$ |
|---|---|---|---|---|---|---|---|---|---|---|---|---|---|---|---|---|---|
| **Type** | A | | B | B | B | C | C | A | B | B | A | C | C | C | A | A | C |
| AMF | | $\lambda(nm)$ | | | | | | | | | | | | | | | |
| **2** | 0.04 | **870** | | | | | 0.11 | 1.29 | 0.18 | 0.62 | 0.52 | 0.43 | 1.29 | 1.46 | 0.66 | 0.06 | 0.67 |
| **4** | 0.19 | **1020** | 0.02 | 0.04 | 0.01 | 0.04 | 0.10 | 0.33 | 0.04 | 0.07 | 0.14 | 0.14 | 0.34 | 0.41 | 0.17 | 0.11 | 0.20 |
| **8** | 0.79 | **1640** | | | | | 0.09 | 0.17 | 0.06 | 0.06 | 0.13 | 0.08 | 0.19 | 0.26 | 0.12 | 0.06 | 0.14 |

## 4 Results

The PYR-ILIOS TOA SSI results are obtained by averaging the $E_\lambda(0)$ obtained by the Langley-plot method for the 12 half-days that satisfied the data selection criteria detailed in Section 2.8. PYR-ILIOS and other space-borne and ground-based instruments datasets described in the Introduction are compared to the SOLAR-ISS(IR) from Meftah et al. (2017) in Figure 3.

The mismatch between the PYR-ILIOS and IRSPERAD dataset, varies between $2\%$ and $4.5\%$ in the central wavelength range between $1.0\mu m$ to $1.8\mu m$, attaining $5\%$ in the $2.1\mu m$ window and peaking to a maximum of $6\%$ in the $1.5\mu m$ and $2.2\mu m$ windows. Except for the shorter wavelengths ($\lambda < 900nm$) region, uncertainties do not explain the observed mismatch between both.The higher disagreement is observed in the far end of the spectrum, with discrepancies of up to $13\%$ between CAVIAR2 and ATLAS3 and SORCE. Below $1.3\mu m$ all the datasets are compatible within the uncertainties bars.

## 5 Discussion

The difference observed between IRSPERAD and PYR-ILIOS is not explained by the uncertainties of both datasets. An atmospheric bias is not considered, because MLO and IZO are world reference sites for the determination of extra-terrestrial constants (Shaw, 1976; Kiedron and Michalsky, 2016; Toledano et al., 2018) and the atmospheric perturbations in ground-based SSI measurements are negligible (Elsey et al., 2017; Bolsée et al., 2016; Weber, 2015). By carrying the new PYR-ILIOS experiment, we unveiled a defect of fixation of the focusing lens. Due to the fact that the instrument was moved between the IRSPERAD pre-campaign relative calibration (May 31 2011) and the start of the Sun measurement campaign (June 1 2011 onwards), the effect of the lens' eventual movement was not considered and therefore not monitored; this defect likely biased the SSI obtained during the IRSPERAD campaign in a non-reproducible way. This defect was detected and corrected for the PYR-ILIOS campaign and the relative calibration strategy adapted to identify possible similar issues: the instrument was installed, powered on and the lamps were measured; the solar measurements began immediately afterwards, without displacing nor powering off the instrument. The PYR-ILIOS relative calibration procedure highlights the importance of ground-based pre-campaign instrument's response monitoring with secondary standards. Additionally it justifies the choice of PYR-ILIOS as a

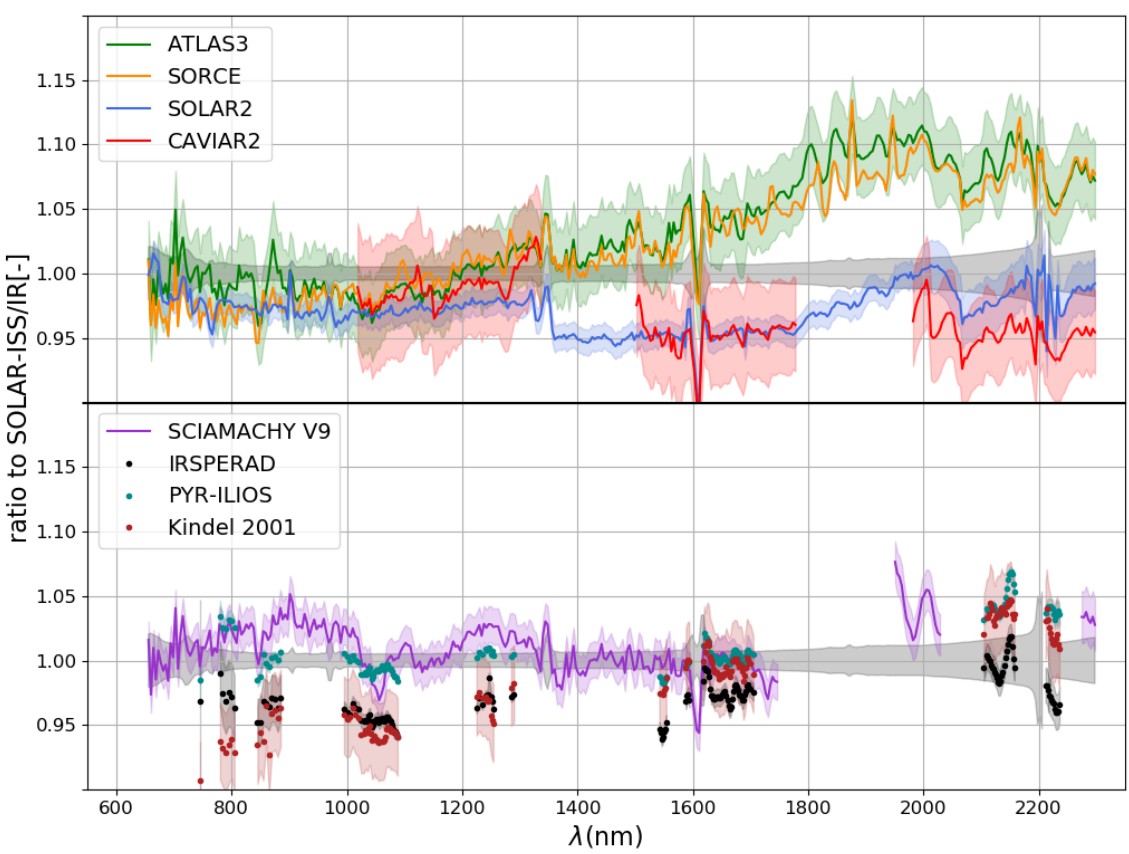

**Figure 3.** Ratio of ground-based and space-borne spectra, relative to SOLSPEC-ISS(IR). Uncertainty at $\pm 1\sigma$ are represented by the shaded areas.

more reliable measurement, than IRSPERAD, due to the higher confidence in the traceability of the instrument's calibration to the blackbody primary standard.

In the higher disagreement region around $1.6\mu m$, the most recent data versions of SOLAR/SOLSPEC and SCIAMACHY instruments, SOLSPEC-ISS and SCIAMACHY V9, respectively, as well as PYR-ILIOS converge to an intermediate level, between SOLAR2 and ATLAS3. This convergence is also observed for longer wavelengths: in the $2\mu m$ region PYR-ILIOS and Kindel are in reasonable agreement, while the level of the two SCIAMACHY V9 adjacent bands ($1.9\mu m - 2.05\mu m$ and $2.2\mu m - 2.4\mu m$) suggests that it also is in agreement with the two ground-based datasets; on the other hand, in this region, both data versions of the SOLSPEC/SOLAR still retain the $8\%$ difference to ATLAS3 and SORCE. A rerun of the measurement campaign at IZO would be crucial to understand the observed discrepancy between PYR-ILIOS and IRSPERAD datasets. Data from SORCE sucessor, TSIS on board ISS since December 2017, is expected to further increase the understanding of the SSI in the NIR.

*Data availability.* The PYR-ILIOS NIR SSI dataset can be downloaded at ftp://ftp-ae.oma.be/dist/PYRILIOS_NIR_SSI/

*Acknowledgements.* The authors which to thank the staff of the Mauna Loa Observatory for kindly supporting the campaign and especially Paul Fukumura-Sawada of the NOAA Earth System Research Laboratory. We thank Brent Holben, PI of MLO AERONET site, for his efforts in establishing and maintaining the MLO site. The authors acknowledge the support from the Belgian Federal Science Policy Office (BELSPO) through the ESA-PRODEX program (contract 4000110593 extension of PEA for 2016-2017) and the funding of the Solar-Terrestrial Centre of Excellence (STCE).

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

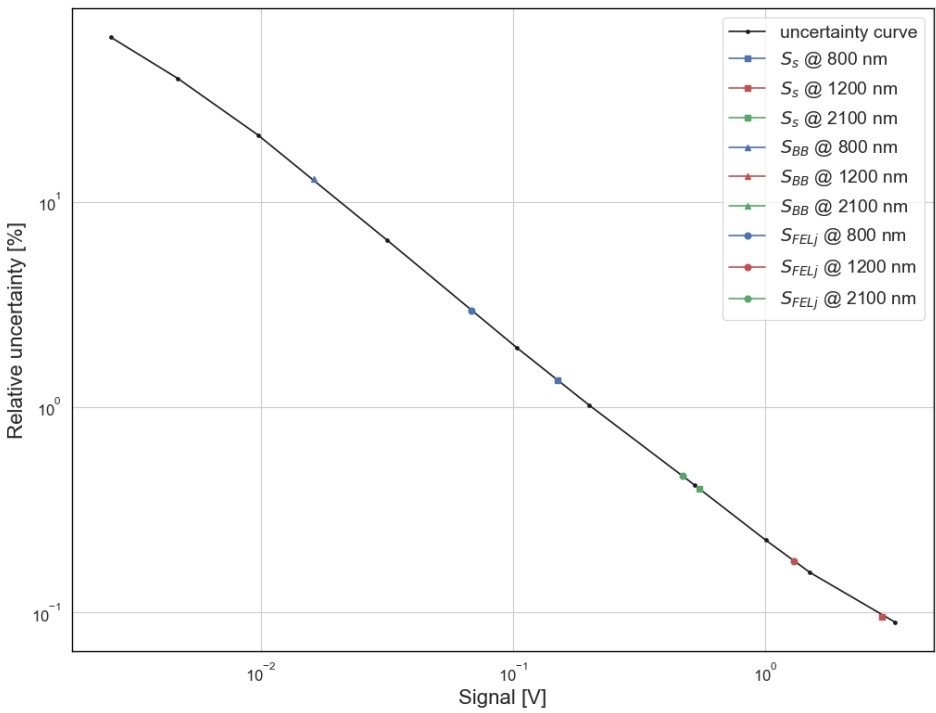

**Figure A1.** Spectrometer's uncertainty curve as a function of the measured signal as determined in laboratory. For reference, the uncertainty values for Solar, blackbody and lamp signals at specific wavelengths are also shown.

Woolliams, E. R., Fox, N. P., Cox, M. G., Harris, P. M., and Harrison, N. J.: Final report on CCPR K1-a: Spectral irradiance from 250 nm to 2500 nm, Metrologia, 43, 02 003, http://stacks.iop.org/0026-1394/43/i=1A/a=02003, 2006.

# Appendix A:  Appendix A

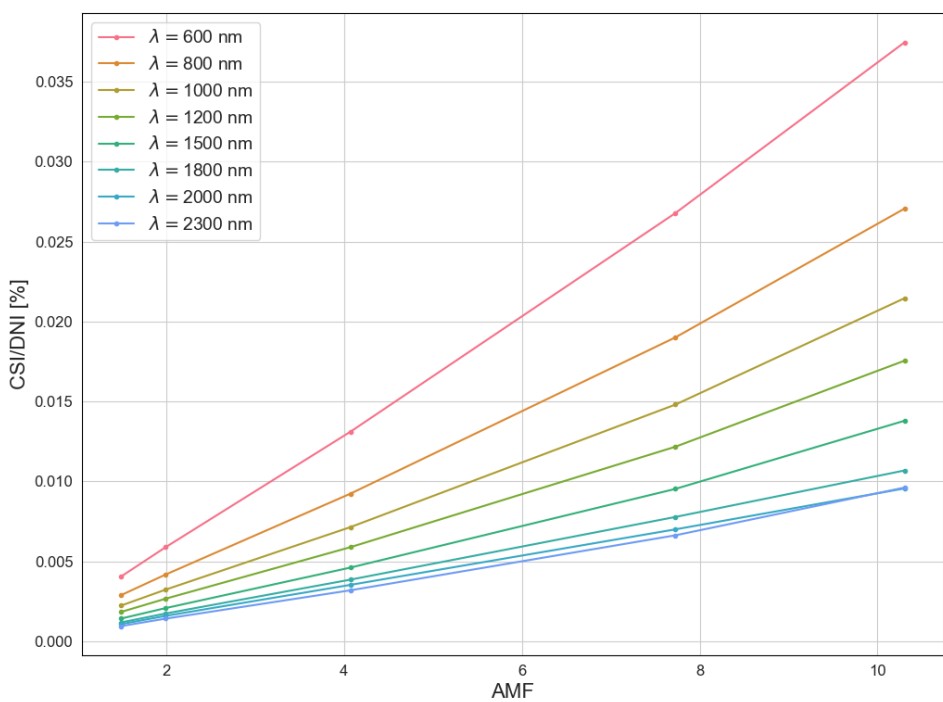

**Figure A2.** Modelled percentage of circumsolar irradiance relative to normal direct irradiance, entering the detector as a function of wavelength and AMF. Circumsolar irradiance has a negligible effect on the measured irradiance even for the highest circumsolar conditions (shorter wavelengths and high AMF).