# Peer review of "Metrology of the Solar Spectral Irradiance at the Top Of Atmosphere in the Near Infrared Measured at Mauna Loa Observatory: The PYR-ILIOS campaign"

_Atmospheric Measurement Techniques, 2018_

## Referee Comment (RC1) · Anonymous Referee #1 · 7 Jul 2018

The manuscript describes measurements of the near infrared solar irradiance from the surface and its extrapolation to top-of-the-atmosphere (TOA) solar irradiance using the Langley-plot technique. The experiment was performed at the high-altitude Observatory in Mauna-Loa, Hawaii and is a follow-up experiment to a similar one performed at Izaña, Tenerife, Canary Islands. The resulting TOA irradiances are compared to space-based measurements of solar irradiance, and a rudimentary uncertainty analysis is provided. The paper provides an important dataset to try to resolve observed discrepancies in space-based solar irradiances in the near infrared range. The advan-

tage of ground-based measurements is the possibility of characterising and calibrating the instrument before, during, and after the measurements in order to validate the measurements.

However the paper needs some modifications before it is ready to be published, as described below.

Minor comments:

- As implied in the title, the paper would like to provide a validated dataset with well defined uncertainties to serve as reference for solar irradiance in the NIR. For that to be accepted, a comprehensive uncertainty budget would be desired, following the traditional terminology used in the metrological community. For example signal to noise is not the most important source of uncertainty, it is in fact quite straightforward, and can be reduced by increasing the sampling, a fact the authors fail to mention (Type a uncertainty). What about the nonlinearity, which could affect the measruements from the differences in intensity between the lamp sources used for the calibration and the solar irradiance? Temperature stability, flat field of the entrance optics, detector stability on diurnal time scales, all could affect the measurements. To be more convincing, the authors might want to be more thorough, since as they claim the measurements performed with the same instrumentation in Izana and Maunaloa differ by more than their estimated uncertainties, indicating that possibly some systematic source of uncertainty was not taken into account. At the end of Section 3.4 I suggest to include an uncertainty table listing the uncertainty components, the type of uncertainty, coverage probability, degree of freedom, if it is simple or expanded uncertainties.

- The airmass calculation used in the paper (see page 6, Figure 1 and later) is based on the Kasten and Young Formula, which assumes a standard US atmosphere. At large airmasses, the profiles of relevant atmospheric species such as water vapour become relevant, and especially at airmass 8 the difference between the profiles of the standard US atmosphere and the one at MaunaLoa could add a discrepancy affecting

the langley-plot retrieval. Could the authors discuss this possible source of uncertainty?

- The paper is missing a conclusion. For example, as a reader I would expect to know if the original problen, namely the observed discrepancies between spectra, is now resolved with this ground-based measurement, what else is needed, and a final outlook.

- The authors might have not been aware of it, but recently a paper has been published in ACPD which compares the quality of langley-plot retrieved solar irradiances between MaunaLoa and Izana. I think the authors should refer to this paper in their introduction and when discussing the Langley-plot technique. (Toledano et al., ACP-2018-430)

technical comments and corrections:

P1, line 16, a reference is missing (?)

P1, line 17 evolved

P2, l 16 : Mention the paper of Toledano et al., here for example

P3, l 15 and where it is used : entrance slit is the correct term

P3, eq. 1: As it is the equation is not correct: the TOA irradiance cannot be divided by the square of the Sun-Earth Distance. It probably should be the squared ratio between the standard Earth-Sun distance and the actual distance at the time of measurement.

P3, l29. change I to I0 for the symbol of top of atmosphere

P4, lines 5-8. The argument needs to be modified: The problem is not gas absorption per se, but the saturation of individual absorption bands, in combination with the resolution of the instrument. For example, in the visible region, the ozone absorption from the Chappuis band does not affect the Langley-retrieval as long as the ozone is stable during the measurements.

P4, l 24, change T to TBB

P4, l 26. Please add a reference for the quoted uncertainties of the emissivity, aperture and distance. The distance uncertainty might be relevant to the calibration performed at PTB with their system, but I doubt that the distance between the entrance optic of the solar spectroradiometer and the Blackbody aperture can be determined to this uncertainty, considering that it is not only a geometrical measurement that is relevant, but the distance between the optical reference planes of both elements (entrance optic and aperture of BB).

P5, l 22-24. The Aeronet network is repeated twice, please remove one instance of the two.

P6, l 7 the the

P 7, Figure 2. Could the combined uncertainty be added to this plot?

P7, Section 3. See earlier comments on the uncertainty treatment.

P8, l 16-17. The uncertainties of the Lamps are not uncorrelated (for example the blackbody is always the same) and can therefore not be combined and divided by sqrt(N-1).

P8, Section 3.3 Here, the atmospheric profiles need to be considered for the AMF calculation and the resulting uncertainty when using a US standard atmosphere. Furthermore, the refraction becomes critical at these large solar zenith angles, and will also add to the uncertainty.

P9, l 23 change "error bars" to "uncertainty"

P9, lines 21-28. I find the discussion partly confusing because some of the spectra are compared with the PYR-ILIOS, but others are compared between each other, and it is not clear what is the result of this paper and what could be moved into the introduction. Could it be rewritten?

P10, Figure 3: What do the shaded areas represent?

P10, l 2, change "error bar" to "uncertainty"

P10, l. 8; I would rephrase this part. It was not "impossible to be monitored", but at the time you did not consider this as a potential source of uncertainty.

P15, Figure A1. It is not clear how this figure supports the paper, and specifically the uncertainty estimation? If it is random noise, then additional sampling would improve the uncertainty (Type-A uncertainty). The x-axis is in volt, but it should be solar irradiance or wavelength to understand the relationship with Figure 2.

P16, caption (and possibly at other places in the manuscript: Please be careful to use the terms shorter or longer wavelengths, instead of upper and lower.

P17, Table A1. I assume AM refers to the morning? Could it be removed from the first column and placed in the caption?

---

## Referee Comment (RC2) · Anonymous Referee #2 · 16 Sep 2018

Summary: This paper presents results from a new field campaign deployment for measuring the solar spectral irradiance in the NIR at Mauna Loa Observatory, based on improvements from the previous campaign, IRSPERAD. The paper describes the use of a spectrometer that has been calibrated, before during and after deployment, tracking the sun over 8 mornings. Particular attention is put into error analysis of the blackbody and tracking the spectrometer variability by using multiple NIST traceable lamp standards. Enhancements from previous field campaign measurements include resolution of instrument issues and a relative calibration strategy.

[Figure]

General Comments: This paper is well written and concise. This paper is valuable and scientifically interesting as it adds another set of measurements to identify the solar NIR spectra, which is currently contested. This paper is recommended for publication with major revision. The uncertainty analysis may not be sufficient to link the measurements to the conclusions, especially as it pertains to the actual Langley regression. Unless the issues addressed below under 'Major Issue' the sentence on line 1-2 of p. 11 should be revised or omitted:" Additionally it justifies the choice of PYR-ILIOS as the more reliable measurement due to the high confidence of the traceability of the instrument's calibration to the blackbody primary standard."

Major issue: The Langley plot example shows a regression of only 4 points (5 in the case of the shortest wavelength), which seems very low. This poses the question of what type of confidence is obtained from the linear fit, and the related uncertainty of the extrapolation to the intercept at origin. Since only 8 days are selected, this does pose the question if the omission or selection of certain Langley regression would influence the resulting values, while the selection criteria seems large ($R^2 > 0.9$, and AOD variation of 10%). Permitting AOD variations within a Langley extrapolation of up to 10% may result in systematic variation of the intercept at origin calculation by up to 10%. In addition to extra care on the Langley regression of few sampled, with increased uncertainty characterization and confidence interval of the intercept at origin, it may be best for the author to include all points from the 8 days for selected wavelengths to help guide the reader in the selection of the Langley extrapolation, and for future reproducibility. In addition, the refined Langley regression method may be better suited here to account for various issues related to varying airmass factor for Aerosol as opposed to Rayleigh scattering (see Schmid and Wehrli, 1995). More advance linear regression of the Langley extrapolation method may also be needed for the days where the Langley-plot has the lower bounds of $R^2$, see description by Shinozuka et al. (2013)

Minor issues: - Identification or description of the flatness off the FOV of the instrument,

within the 0.1° sun-tracker accuracy, and stability through temperature variations when exposed to ambient variations at the mountain top. - Potential detector non-linearity not addressed for actual measured irradiance values (only using a 2 point calibration values which may not represent irradiance levels sampling during Langley regression). - In the calculation of the uncertainty on the determination of TOA irradiance, the AMF error uncertainty should not be considered zero, variation of the distribution of the aerosol, or some trace gases affecting the column within the atmospheric column as compared to pure Rayleigh scattering AMF, should at the very minimum give some indication of the potential error in AMF calculation. At the very least, error propagation of the pressure measurement's uncertainty to the AMF calculation should be made to determine the AMF uncertainty. Variability of the AMF during measurement time period should also be determined. The Meeus derivation of solar zenith angle is expected to be an overestimate of the apparent solar zenith angle (because of atmospheric refraction), where the Duffett-Smith (1988) may provide a more accurate position.

Here are some specific points to be addressed: p. 1, line 16 – reference missing. p.1, line 17 – 'evoluted' typo? And this sentence requires some reference to prove the point that consensus on the absolute NIR level is still to be achieved. p.2 line 23 – nonsensical sentence: 'it is nowadays the instrument that measured farther the SSI in the NIR.' p.6 line 9 – 'consulted in A1' should be 'consulted in Table A1' Figure 3 – is difficult to see, color choices and symbols should be revised. Table A1: formatting of this table is confusing. The 'AM' moniker should be removed and included in the table description.

References: P. Duffett-Smith, "Practical astronomy with your calculator," 3rd ed. 1Cambridge U. Press, Cambridge, England, 19882. Schmid, B., and C. Wehrli (1995), Comparison of Sun photometer calibration by use of the Langley technique and the standard lamp, Appl. Opt., 34(21), 4500–4512. Shinozuka, Y., Johnson, R. R., Flynn, C. J., Russell, P. B., Schmid, B., Redemann, J., Dunagan, S. E., Kluzek, C. D., Hubbe, J. M., Segal-Rosenheimer, M., Livingston, J. M., Eck, T. F., Wagener, R., Gregory, L., Chand,

D., Berg, L. K., Rogers, R. R., Ferrare, R. A., Hair, J. W., Hostetler, C. A. and Burton, S. P.: Hyperspectral aerosol optical depths from TCAP flights, J. Geophys. Res. Atmos., 118(21), 12180–12194, doi:10.1002/2013JD020596, 2013.
* * *

---

## Author Response (AR1)

**Response to the reviewers**

Response to referee #1:

The authors acknowledge the valuable comments of the referee. The Toledano reference, unknown to us, comes in very useful.
Regarding the comments on the stability of the detector, some of the issues raised have been clarified in the text.

**Minor comments**

- Signal to noise is not a source of uncertainty per se. It is a quantity intrinsic to the spectrometer (entrance optics + light dispersing system + detection), determined in laboratory and used as a look-up-table for the uncertainty of a measured signal given its intensity. As the use of SNR for uncertainty calculation could be confusing we opted to remove the reference to SNR and update figure A1 to plot uncertainty instead.
There was no application of sampling increasing for signal to noise reduction.
- Non linearity issues: the inverse square law was verified in laboratory; this is stated in new section 2.5
- The flat-field of the instrument was measured during the ground based campaign at MLO. The telescope was interfaced with the sun tracker body through an angular fine-tuning mechanism; this allowed us to establish a precise parallelism between the sun tracker detector and telescope sun-facing surfaces. This mechanism permitted to precisely depoint the telescope relative to the solar tracker and thus the Sun for a series of angles, for two perpendicular directions. The results are shown in supplementary material. Green and blue markers represent the two perpendicular directions of depointing. The dashed lines represent the Sun tracking accuracy, < 0.01°, provided by the manufacturer. The response of the detector is flat within these limits as shown by both curves.
- The temperature sensitivity of the spectrometer was thoroughly determined in laboratory as referred in Bolsée 2014 (sec 2.2.3). Due to logistics constraints, during the Izaña campaign the spectrometer had to be placed outdoors. Its temperature reached occasionally 40°C (nominal temperature set point is 24.7 °C), requiring a considerable correction to the signal. The situation was different during the campaign at MLO: the spectrometer was placed indoors and its temperature was constant within 0.1°C. Referred in section 2.5
- Regarding detector stability on diurnal time scales, this monitoring is not feasible. Detector response is monitored every week as explained in section 2.5.
- A table listing the individual uncertainty is now included at the end of section 3.5.

Air masses were calculated using the Schmid and Wehrli approach, based on the Kasten&Young algorithm; however this wasn't explained but it is now the case in the section 2.2.. Also, their method to estimate the uncertainty in the AMF calculation is now included in the new section 3.5. (Estimation of air mass factors uncertainties).

We now explain at the beginning of the second paragraph that  the discussion dedicated to the NIR SSI debate is not closed. A sentence has been added proposing a solution to understanding the discrepancies observed between PYR-ILIOS and IRSPERAD.

Reference added.

**Technical comments and corrections**

**P1,l16**: Reference now showing correctly

**P1,l17**: Corrected

**P2,l16**: Reference inserted

**P3,l15**: Corrected 2 occurrences

**P3,eq1**: Corrected

**P4,l5-8**: In order to avoid confusion w.r.t. to the applicability of Langley-plot to gas absorbing spectral regions, sec 2.2. was rewritten.

**P4,l24**: Done

**P4,l26**: Reference added. This comment is particularly valuable due to the fact that the meaningful distance to be considered here is the effective distance between the blackbody and the optical centre of the telescope. This distance is a sum of two different distances each determined with different uncertainties.

**P5,l22-24**: Done

**P6,l7**: Done

**P7, fig2**: This is already the case. The combined uncertainty is represented by $u(E_0)$. To make this clearer, caption was updated accordingly.

**P7, sec3**:

**P8, l16-17**: The uncertainty on the relative calibration factor, K, is obtained by applying LPU to eq. 5, with each lamp having its own independent uncertainty on the measured signal. On lines 10 and 11 is only mentioned that LPU was applied to E. This could cause misunderstanding and was updated to state that LPU is applied to all the factors in eq 7. The last sentence was deleted in order to clarify the text.

**P9, l23:** Done

**P8, Sec.3.3:** The answer to the first remark is included in the 2nd minor comment. Although not referred in text, the solar zenith angle calculated with Meeus [1998] algorithm was corrected for atmospheric refraction, according to Bennet [1982]. This is now stated in the text.

**P9, l23:** Done

**P9, l21-28**: The objective of this campaign was to provide a new input dataset for the solar TOA NIR level. As a rerun of the campaign of 2011 performed with the same instrumentation, the first sentences of Results are naturally dedicated to comparing IRSPERAD and PYR-ILIOS. The choice of SOLSPEC-ISS(IR) is to facilitate the comparison between all the datasets. It would be a priori more suitable to choose PYR-ILIOS as reference, however its limited spectral coverage would weaken the analysis between space borne datasets. Results section was slightly rewritten to increase clearness.

**P10, fig3:** shaded areas represent 1-sigma uncertainties; caption updated

**P10, l2:** Done

**P10, l8:** Done

**P15, fig A1:** We understand that the way that the concept of SNR was introduced could lead to confusion on the understanding on how the uncertainty on the signal is determined. Section 3.1 was rewritten and Fig A1 converted to uncertainty as a function of the signal.
**P16, caption:** Done
**P7, Tab.A1:** corrected

**Response to referee #2:**

The authors acknowledge the inputs of the referee which added value to the quality of the manuscript, namely concerning the quantification of the uncertainties in air mass factors.

**Major issues**

- **General comment:** the sentence on lines 1-2 of p.11 has been rewritten as it was intended to only compare IRSPERAD (Izana 2011) and PYRILIOS (MLO 2016) and declare PYIRILIOS as the more reliable of the two and not of all NIR datasets.
- **Langley plots selection:** section 2.7 was updated taking into account the reviewer comments: The criterion for the selection of Langley plots (max 10% variation during half-day) was replaced by an analysis of the sensitivity of the Langley method to the aerosol optical depth (AOD) as measured by the AERONET instruments. The only remaining data pre-reduction criterion is the selection for cloudless clear sky (half-) days. The removal of this criterion allowed us to increase not only the number of half-days available (8 to 12) but also to work with lower AMFs. The results of the analysis of this sensitivity are presented on section 3.6. Figure 1 updated accordingly.
- **Air mass calculations:** The Schmid and Wehrli approach for the calculation of air masses was already being used in this data treatment (but not stated); their method to estimate the uncertainty in the AMF calculation is now included; section 2.2. was rewritten and section 3.5. (Estimation of air mass factors uncertainties) was added.

**Minor issues**

- The flat-field of the instrument was measured during the ground based campaign at MLO. The telescope was interfaced with the sun tracker body through an angular fine-tuning mechanism; this allowed us to establish a precise parallelism between the sun tracker detector and telescope sun-facing surfaces. This mechanism permitted to precisely depoint the telescope relative to the solar tracker and thus the Sun for a series of angles, for two perpendicular directions. The results are shown in supplementary material . Green and blue markers represent the two perpendicular directions of depointing. The dashed lines represent the Sun tracking accuracy, < 0.01°, provided by the manufacturer the response of the detector is flat within these limits as seen by both curves.
- The linearity of the instrument was measured during the PTB calibration. It was previously referred in page 5 lines 2&3 that the blackbody two temperature set points are used to verify the linearity of the spectrometer. This can be misleading as this was just an extra verification of the linearity. The linearity was verified using a dedicated experimental setup in which the entrance optics was moved away from a stable 200W lamp for a series of known distances; the measured distance-irradiances data points were successfully fitted to an inverse square law function.

- See last item of Major issues
- Although not previously referred in text, the solar zenith angle calculated with Meeus [1998] algorithm was corrected for atmospheric refraction, according to Bennet [1982]. This is now stated in the text.

**Specific points**

- **P1,l16:** reference added
- **P1,l17:** done
- **P2,l23:** done
- **P6,l9:** no table A1 anymore
- **Figure 3: figure updated**
- **Table A1:** no table A1 anymore

**List of changes**

The more substantial changes on the manuscript are related to the working Langley plots AMF working range and the AMF sensitivity

1) The extension of the Langley plots calculations for AMF < 2 led to the following changes:
   a. Recalculation of the $E_0$ values: update to figures 1 and 3; no substantial deviations to previous $E_0$ values were found susceptible of altering results and conclusions.
   b. Recalculation of the final uncertainty on the $E_0$ values: the substantial increase in the Langley plot points (from 5/6 to more than 15 per wavelength) significantly decreased the uncertainty on $E_0$.

2) A study of the sensitivity of the $E_0$ to the AMF calculation led to the following changes:
   a. The suppression of aerosol stability as Langley plot selection criteria (Sec. 2.7); this led to the increase from 8 to 12 usable Langley plot half days and to the suppression of Table A1.
   b. The inclusion of the estimation of the uncertainty in calculating AMF (Sec. 3.5) and is now included in the $E_0$ uncertainty calculation.
   c. The inclusion of the Langley plot $E_0$ sensitivity estimation to real daily aerosol variations (Sec. 3.6) and now included in the $E_0$ uncertainty calculation.

3) Other substantial changes
   a. Recalculation of the blackbody to entrance optics distance (Sec. 2.4).
   b. Inclusion of uncertainty listing table: Table 1.
   c. Rewritten of the Results and conclusions according to the suggestions of the reviewers.
   d. In order to address reviewers' questions, concerning the determination of the detector's linearity, flat field and stability a new section, 2.5 Radiometric characterization, was added.

Figure 2 was updated accordingly to the changes in 1 b), 2 b), 2 c) and 3 a)

Other changes

- Rewritten of the Langley plot method description (Sec. 2.2).
- Change of figure A1 to express uncertainty instead of signal to noise ratio as a function of signal.
- Clarification on the indexes $i$ notation in Sec. 2.6.
- Correction of the description of the factor $D$ of equations 1 and 2.

Minor changes:

- Consistency of the utilization of $E$ instead of $I$ for irradiance.
- Consistency between Instruments/datasets and figures legends.
- Correction of some bibliographical items.

- Update of figures general readability and captions.

[revised manuscript text omitted]

---

## Referee Report (RR1)

**2nd Review of "Metrology of the Solar Spectral Irradiance at the Top Of Atmosphere in the Near Infrared Measured at Mauna Loa Observatory: The PYR-ILIOS campaign" by Pereira et al.**

*Summary:*

This revision addressed all major issues of the original manuscript and seems to make this publication stronger. That said, clarifications on the added details are recommended before publication.

*Remaining issues:*

Section 2.5:

- This section could use some refinement in the presentation.
- Point 1 in this section should describe by how far in angular degrees the field of view is 'flat' without deviation from the highest transmission value. In order words, describe the supplemental figure that was sent out. The figure does show 1% change in one direction at 0.3°, without finer resolution of this measurement, this detail should be mentioned.
- Point 3 describe the verification of the linearity of the detector, similarly describe, but with a different method in line 21 of p.5. These two descriptions should either be combined and contrasted as it is confusing. From the author's response, one is a determination of non-linearity, the other is a verification.

Figure 2.:

- Why is the combined relative uncertainty in TOA SSI lower than its governing values, especially at wavelengths lower than 1000nm?

Section 3.6:

- "measured AOD profile", here what does a 'profile' indicate? Vertical profile, or diurnal profile? Please clarify.
- Equation seems to indicate that the Rayleigh optical depth is not time variant, which should be considered if not already.
- Line 12, p.11, "corresponding uncertainty amounts to" It is unclear how the averaged bias values lead to those values of the uncertainty. Additionally, since the values can be both positive and negative, does it make sense to report the root mean square error instead of the average?

Table 1.:

- There are remaining notations of 'I' to denote I believe Irradiance instead of the the term 'E' used everywhere else in the manuscript.

---

## Author Response (AR2)

**2nd Review of "Metrology of the Solar Spectral Irradiance at the Top Of Atmosphere in the Near Infrared Measured at Mauna Loa Observatory: The PYR-ILIOS campaign" by Pereira et al.**

*Summary:*

This revision addressed all major issues of the original manuscript and seems to make this publication stronger. That said, clarifications on the added details are recommended before publication.

*Remaining issues:*

Section 2.5:

- This section could use some refinement in the presentation.

- Point 1 in this section should describe by how far in angular degrees the field of view is 'flat' without deviation from the highest transmission value. In order words, describe the supplemental figure that was sent out. The figure does show 1% change in one direction at 0.3°, without finer resolution of this measurement, this detail should be mentioned.

The coarse angular sampling of the measurement (15') doesn't indeed allow to finely describe the flat-field at the top of the curves. The mismatch between curves and the drop of 1% at 0.3° in one of the 4 directions is most probably due to an incorrect normalization of the Sun signal with AMF and/or atmospheric perturbations.

Given the flat-field curves we estimate the instrument pointing insensitivity to be ~0.05°, covering the 0.01° pointing accuracy given by the manufacturer.

Point 1 was rewritten accordingly.

- Point 3 describe the verification of the linearity of the detector, similarly describe, but with a different method in line 21 of p.5. These two descriptions should either be combined and contrasted as it is confusing. From the author's response, one is a determination of non-linearity, the other is a verification.

To eliminate this source of confusion, the mention to the verification of linearity with the blackbody was deleted.

Figure 2.:

- Why is the combined relative uncertainty in TOA SSI lower than its governing values, especially at wavelengths lower than 1000nm?

Thanks to this comment some errors were detected in this plot: an old (incorrect) version of the uncertainty in R (although the correct u(R) one was plotted) was used to calculate the uncertainties in E and $E_0$; this is now corrected: all u(E) values are at least higher than u(R) which is in mathematical agreement with eq. 7.

Figure 2 and Table 1 are corrected accordingly.

Now, concerning the comment itself, I assume the referee is referring to the u($E_0$) being lower than u(E):

In the first version of the article the opposite happened [i.e. u($E_0$) > u(E)] which made sense to us from a LPU point of view; we were working with a [2,8] AMF range back then, meaning we had 4-5 points for the Langley-plot. After the reviewers comments we extended the AMF range to [1,8] increasing the number of Langley-plot points to around 20. When the u($E_0$) was recalculated we verified that u($E_0$) < u(E) which was attributed to the increasing of the number of Langley plot points having a sort of statistical effect on the determination of the intercept at origin to such an extent that it becomes lower than the uncertainties on the governing values; this was also verified by the Monte Carlo method, referred in section 3.3.

This is our qualitative understanding of this observation.

Section 3.6:
- "measured AOD profile", here what does a 'profile' indicate? Vertical profile, or diurnal profile? Please clarify.

Clarification done, we meant a diurnal profile.

- Equation seems to indicate that the Rayleigh optical depth is not time variant, which should be considered if not already.

The calculation of the Rayleigh optical depth takes into account MLO's diurnal atmospheric pressure variation.

- Line 12, p.11, "corresponding uncertainty amounts to" It is unclear how the averaged bias values lead to those values of the uncertainty. Additionally, since the values can be both positive and negative, does it make sense to report the root mean square error instead of the average?

Our intention was to first present the averaged bias as an indication of the deviation that can be induced at the determination of the intercept at origin by not considering an aerosol dynamic atmosphere; hence the small analysis of this bias in the sentence that followed.

To calculate the associated uncertainty we make the assumption that the determined, $E_0^{LP}$ and $E_0^*$, are the extremes of a rectangular distribution that contains the true value $E_0$, for which the uncertainty is $| E_0^{LP} - E_0^*|/2\sqrt{3}$. This last calculation step was now included, the mention to bias in line 10 removed for clearness as well as rephrasing of the corresponding sentence.

Table 1.:
- There are remaining notations of 'I' to denote I believe Irradiance instead of the the term 'E' used everywhere else in the manuscript.

Thanks for the warning. Done.

[revised manuscript text omitted]